# Learning Bayesian Networks with Low Rank Conditional Probability Tables

**Adarsh Barik**
Department of Computer Science
Purdue University
West Lafayette, Indiana, USA
`abarik@purdue.edu`

**Jean Honorio**
Department of Computer Science
Purdue University
West Lafayette, Indiana, USA
`jhonorio@purdue.edu`

## Abstract

In this paper, we provide a method to learn the directed structure of a Bayesian network using data. The data is accessed by making conditional probability queries to a black-box model. We introduce a notion of simplicity of representation of conditional probability tables for the nodes in the Bayesian network, that we call "low rankness". We connect this notion to the Fourier transformation of real valued set functions and propose a method which learns the exact directed structure of a 'low rank' Bayesian network using very few queries. We formally prove that our method correctly recovers the true directed structure, runs in polynomial time and only needs polynomial samples with respect to the number of nodes. We also provide further improvements in efficiency if we have access to some observational data.

## 1   Introduction

**Motivation.**   Real-world systems are made of large number of constituent variables. Understanding the interactions and relationships of these variables is key to understand the behavior of such systems. Scientists and researchers from many domains have been using graphs to model and learn relationships amongst variables of real-world systems for a long time. *Bayesian networks* are one of the most important classes of probabilistic graphical models which are used to model complex systems. They provide a compact representation of joint probability distributions among a set of variables.

**Related work.**   Learning the structure of a Bayesian network from observational data is a well known but an incredibly difficult problem to solve in the machine learning community. Due to its popularity and applications, a considerable amount of work has been done in this field. Most of these work use *observational data* to learn the structure. We can broadly divide these methods in two categories. The methods in the first category use score maximization techniques to learn the DAG from observational data. In this category, there are some heuristics based approaches such as Friedman et al. (1999); Tsamardinos et al. (2006); Margaritis and Thrun (2000); Moore and Wong (2003) which run in polynomial-time without offering any convergence/consistency guarantee. There are also some exact but exponential time score maximizing exact algorithms such as Koivisto and Sood (2004); Silander and Myllymäki (2006); Cussens (2008); Jaakkola et al. (2010). The methods in the second category are independence test based methods such as Spirtes et al. (2000); Cheng et al. (2002); Yehezkel and Lerner (2005); Xie and Geng (2008).

There have also been some work to learn the structure of a Bayesian network using *interventional data* (Murphy, 2001; Tong and Koller, 2001; Eaton and Murphy, 2007; Triantafillou and Tsamardinos, 2015). Most of these works first find a Markov equivalence class from observational data and then direct the edges using interventions. Unfortunately, the first step of finding Markov equivalence class

remains NP-hard (Chickering, 1996). Hausar and Bühlmann (2012), He and Geng (2008), Kocaoglu et al. (2017) have presented polynomial time methods to find an optimal set of interventions for chordal DAGs. Bello and Honorio (2018) have proposed a method to learn a Bayesian network using interventional path queries with logarithmic sample complexity. However, their method runs in exponential time in terms of the number of parents.

In this paper, our work takes an intermediate path. We do not use pure observational or interventional data directly. Rather, we assume that there exists a *black-box* which answers conditional probability queries by outputting observational data. Our goal is to limit the number of such queries and learn the directed structure of a Bayesian network. We propose a novel algorithm to achieve this goal. We also provide a method to improve our results by having access to some observational data. We intend to measure our performance based on the following criteria. 1. **Correctness -** We want to come up with a method which correctly recovers the directed structure of a Bayesian network with provable theoretical guarantees. 2. **Computational efficiency -** The method must run fast enough to handle the high dimensional cases. Ideally, we want to have polynomial time complexity with respect to the number of nodes. 3. **Sample complexity -** We would like to use as few samples as possible for recovering the structure of the Bayesian network. As with the time complexity, we want to achieve polynomial sample complexity with respect to the size of the network.

**Contribution.** Consider a binary node $i$ of a Bayesian network with $m$ parents. The conditional probability table (CPT) of node $i$ has $2^{m+1}$ entries. This number quickly becomes very large even for modest values of $m$. To handle such large tables while still maintaining the effect of all the parents, we introduce a notion of simplicity of representation of the CPTs, which we call "low rankness". Our intuition is that each CPT can be treated as summation of multiple simple tables, each of them depending only on a handful of parents (say $k$ parents where $k$ is the rank of the CPT). We connect this notion of rank of a CPT to the Fourier transformation of a specific real valued set function (Stobbe and Krause, 2012) and use compressed sensing techniques (Rauhut, 2010) to show that the Fourier coefficients of this set function can be used to learn the structure of the Bayesian network. While doing so, we provide a method with theoretical guarantees of correctness, and which works in polynomial time and sample complexity. Our method requires computation of conditional probabilities from data. We do this by making queries to a black-box. One query consists of two steps. The first step is the selection of variables, i.e., choosing a target variable and a set of variables for conditioning. The second step is to assign specific values to the selected conditioning variables. This process is similar to the process used in Bello and Honorio (2018); Kocaoglu et al. (2017), which consider a particular selection of variables as one intervention. An actual setting of the variables are considered as one experiment. For example, a selection of $k$ binary variables can be assigned $2^k$ distinct values and can be queried in $2^k$ different ways. Our setting is similar to an interventional setting where a selection can be compared to an intervention and an assignment can be compared to an experiment, although our method never queries the $2^k$ distinct values, but a single random assignment instead. Thus, we compare our results to the state-of-the-art interventional methods in Table 1. It should be noted that the number of queries (or experiments in the interventional setting) are a better metric for comparison than the number of selections (or interventions). This is because a selection may involve only one node (Bello and Honorio, 2018) or multiple nodes (in this paper) and thus could hide some complexity of the problem. Furthermore, the sample complexity of the problem depends on the number of queries.

Table 1: Sample and time complexity, number of selections (interventions) and queries (experiments) required for structure learning of binary Bayesian networks. Here $n$ is the number of nodes, $k$ is the maximum size of the Markov blanket. The maximum number of parents of a node is $\mathcal{O}(k)$.

| Algorithms | Sample Complexity | Time Complexity | Selections | Queries |
|---|---|---|---|---|
| Our Work (no observational data) | Blackbox - $\mathcal{O}(nk^3 \log^4 n(\log k + \log\log n))$ | $\mathcal{O}(n^4 k\sqrt{n}\log n)$ | $\mathcal{O}(n)$ | $\mathcal{O}(nk^3 \log^4 n)$ |
| Our Work (with observational data) | Observational - $\mathcal{O}(n)$ Blackbox - $\mathcal{O}(nk^3 \log^5 k)$ | $\mathcal{O}(n^4)$ $\mathcal{O}(nk^4\sqrt{k}\log k)$ | $\mathcal{O}(n)$ | $\mathcal{O}(nk^3 \log^4 k)$ |
| Bello and Honorio (2018) | Interventional - $\mathcal{O}(n^2 2^k \log n)$ | $\mathcal{O}(n^2 2^k \log n)$ | $\mathcal{O}(n^2)$ | $\mathcal{O}(n^2 2^k)$ |
| Kocaoglu et al. (2017) | Interventional - no guarantees | $\mathcal{O}(2^n kn^2 \log^2 n)$ | $\mathcal{O}(\log n)$ | $\mathcal{O}(2^n \log n)$ |

## 2  Preliminaries

In this section, we introduce formal definitions and notations. Let $\mathbf{X} = \{X_1, X_2, \ldots, X_n\}$ be a set of random variables. For a set $A$, $X_A$ denotes the set of random variables $X_i \in \mathbf{X}$ such that $i \in A$. We use the shorthand notation $\bar{i}$ to denote $V \backslash \{i\}$. We define a Bayesian network on a directed acyclic graph $G = (V, E)$ where $V$ denotes the set of vertices and $E$ is a set of ordered pair of nodes, each corresponding to a directed edge, i.e., if $(a, b) \in E$ then there is an edge $a \to b$ in $G$. The parents of a node $i, \forall i \in V$ denoted by $\pi_G(i)$, are set of all nodes $j$ such that edge $(j, i) \in E$. We also define the Markov blanket $\mathrm{MB}_G(i)$ for a node $i$ as a set of nodes containing parents, children and parents of children of node $i$. The nodes with no children are called terminal nodes.

**Definition 1** (Bayesian network). *Let $G = (V, E)$ be a directed acyclic graph (DAG) and $\mathbf{X} = \{X_1, X_2, \ldots, X_n\}$ be a set of random variables such that $X_i$ corresponds to a random variable at node $i \in V, \forall i = \{1, \ldots, n\}$. Let $X_{\pi_G(i)}$ denote the set of random variables defined on the parents of node $i$ in DAG $G$. A Bayesian network $\mathcal{B} = (G, \mathcal{P})$ represents a joint probability distribution $\mathcal{P}$ over the set of random variables $\mathbf{X}$ defined on the nodes of DAG $G$ which factorizes according to the DAG structure, i.e., $\mathcal{P}(X_1, X_2, \ldots, X_n) = \prod_{i=1}^{n} \mathcal{P}(X_i | X_{\pi_G(i)})$ where $\mathcal{P}(X_i | X_{\pi_G(i)})$ denotes conditional probability distribution (CPD) of node $i$ given its parents in DAG $G$.*

We denote the domain of a random variable $X_i, \forall i \in \{1, \ldots, n\}$ by $\mathrm{dom}(X_i)$. The cardinality of a set is denoted by notation $|\cdot|$. A Bayesian network $\mathcal{B} = (G, \mathcal{P})$ on discrete nodes is called a *binary Bayesian network* if $|\mathrm{dom}(X_i)| = 2, \forall i \in \{1, \ldots, n\}$. For discrete nodes, $\mathcal{P}(X_i | X_{\pi_G(i)})$ is often represented as a conditional probability table (CPT) with $|\mathrm{dom}(X_i)| \prod_{j \in \pi_G(i)} |\mathrm{dom}(X_j)|$ entries. In this work, we will only focus on binary Bayesian networks. Next, we introduce a novel concept of rank of a conditional probability distribution for a node of Bayesian network.

**Definition 2** (Rank $k$ conditional probability distribution). *A node $i \in V$ of a Bayesian network $\mathcal{B}(G, \mathcal{P})$ is said to be rank $k$ representable with respect to a set $A(i) \subseteq V \backslash \{i\}$ and probability distribution $\mathcal{P}$ if,*

$$\mathcal{P}(X_i = x_i | X_{A(i)} = x_{A(i)}) = \sum_{\substack{S \subseteq \{i\} \cup A(i) \\ 1 \leqslant |S| \leqslant k, \ i \in S}} Q_S(X_S = x_S), \forall x_i \in \mathrm{dom}(X_i), x_{A(i)} \in \mathrm{dom}(X_{A(i)}) \tag{1}$$

*where $Q_S : \times_{j \in S} \mathrm{dom}(X_j) \to \mathbb{R}$ is a function which depends only on the variables $X_S$. A node $i$ is said to have rank $k$ conditional probability table if it is rank $k$ representable but is not rank $k - 1$ representable with respect to $A(i)$ and $\mathcal{P}$.*

For example, a node $i \in V$ of a Bayesian network $\mathcal{B}(G, \mathcal{P})$ is rank 2 representable with respect to its parents $\pi_G(i)$ and $\mathcal{P}$ if we can write $\mathcal{P}(X_i = x_i | X_{\pi_G(i)} = x_{\pi_G(i)}) = Q_i(X_i = x_i) + \sum_{j \in \pi_G(i)} Q_{ij}(X_i = x_i, X_j = x_j)$, where $\forall x_i \in \mathrm{dom}(X_i), x_j \in \mathrm{dom}(X_j), \forall j \in \pi_G(i)$. It is easy to observe that any node $i \in V$ is always rank $|A(i)| + 1$ representable with respect to a set $A(i) \subset V \backslash \{i\}$ and $\mathcal{P}$. Also, rank $k$ representations for a node $i$ with respect to $A(i)$ and $\mathcal{P}$ may not be unique. We consider real-valued set functions on a set $T$ of cardinality $t$ defined as $f : 2^T \to \mathbb{R}$ where $2^T$ denotes the power set of $T$. Let $\mathcal{F}$ be the space of all such functions, with corresponding inner product $\langle f, g \rangle \triangleq 2^{-t} \sum_{A \in 2^T} f(A)g(A)$. The space $\mathcal{F}$ has a natural Fourier basis, and in our set function notation the corresponding Fourier basis vectors are $\psi_B(A) \triangleq (-1)^{|A \cap B|}$. We define the Fourier transformation coefficients of function $f$ as $\hat{f}(B) \triangleq \langle f, \psi_B \rangle = 2^{-t} \sum_{A \in 2^T} f(A)(-1)^{|A \cap B|}$. Using Fourier coefficients, the function $f$ can be reconstructed as:

$$f(A) = \sum_{B \in 2^T} \hat{f}(B)\psi_B(A) \tag{2}$$

The Fourier support of a set function is the collection of subsets with nonzero Fourier coefficient: $\mathrm{support}(\hat{f}) \triangleq \{B \in 2^T | \hat{f}(B) \neq 0\}$.

## 3  Method and Theoretical Analysis

In this section, we develop our method for learning the structure of a Bayesian network and provide theoretical guarantees for correct and efficient learning. First we would like to mention some technical assumptions.

**Assumption 1** (Availability of Black-box). *For a Bayesian network $\mathcal{B}(G, \mathcal{P})$, we can submit a conditional probability query $BB(i, A, x_A, N)$ to a black-box on any set of selected nodes $i \in V, A \subseteq \bar{i}$ and value $x_A$, and receive $N$ i.i.d. samples from the conditional distribution $\mathcal{P}(X_i | X_A = x_A)$.*

**Assumption 2** (Faithfulness). *The distribution over the nodes of the Bayesian network $\mathcal{B}(G, \mathcal{P})$ induced by $(G, \mathcal{P})$ exhibits no other independencies beyond those implied by the structure of $G$.*

**Assumption 3** (Low rank CPTs). *Each node $i \in V$ in the Bayesian network $\mathcal{B}(G, \mathcal{P})$ has rank 2 conditional probability tables with respect to $\pi_G(i)$ and $\mathcal{P}$.*

Assumption 1 implies the availability of observational data for all queries. This is analogous to the standard assumption of availability of interventional data in interventional setting (Murphy, 2001; He and Geng, 2008; Kocaoglu et al., 2017; Tong and Koller, 2001; Hausar and Bühlmann, 2012). Assumption 2 is also a standard assumption (Kocaoglu et al., 2017; Tong and Koller, 2001; He and Geng, 2008; Spirtes et al., 2000; Triantafillou and Tsamardinos, 2015) which ensures that we only have those independence relations between nodes which come from d-separation. We also introduce a novel Assumption 3 which ensure that CPTs of nodes have a simple representation. In the later sections, we relate this to sparsity in the Fourier domain. We note that there is nothing special about CPTs being rank 2 and our method can be extended for any rank $k$ CPTs.

### 3.1 Problem Description

In this work, we address the following question:

**Problem 1** (Recovering structure of a Bayesian network using black-box queries). *Consider we have access to a black-box which provides observational data for our conditional probability queries for a faithful Bayesian network $\mathcal{B}(G, \mathcal{P})$ with each node $i$ having rank 2 CPT with respect to its parents $\pi_G(i)$ and $\mathcal{P}$. Can we recover the directed structure of $G$ with theoretical guarantees of correctness and efficiency in terms of time and sample complexity?*

We show that it is indeed possible to do. We control the number of samples by controlling the number of queries. We also show that it is possible to further reduce the sample complexity if we have access to some observational data.

### 3.2 Theoretical Result

In this subsection, we state our theoretical results. We start by analyzing terminal nodes.

**Analyzing Terminal Nodes.**    Since terminal nodes do not have any children, their Markov blanket only contains their parents. Furthermore, if the Bayesian network is faithful then for any terminal node $t \in V$: $\mathcal{P}(X_t | X_{\pi_G(t)}) = \mathcal{P}(X_t | X_{\mathrm{MB}_G(t)}) = \mathcal{P}(X_t | X_{\bar{t}})$. Thus, for any terminal node $t \in V$, $\mathcal{P}(X_t | X_{\pi_G(t)})$ can be computed without explicitly knowing its parents. Next, we define a set function which computes $\mathcal{P}(X_t | X_{\pi_G(t)})$. In particular, for an assignment $x_{\pi_G(t)} \in \{0, 1\}^{|\pi_G(t)|}$, we are interested in computing $\mathcal{P}(X_t = 1 | X_{\pi_G(t)} = x_{\pi_G(t)})$. Note that $\mathcal{P}(X_t = 0 | X_{\pi_G(t)} = x_{\pi_G(t)})$ can simply be computed by subtracting $\mathcal{P}(X_t = 1 | X_{\pi_G(t)} = x_{\pi_G(t)})$ from 1. Let $\bar{t}$ denote the set $V \backslash \{t\}$. For node $t$ and a set $A \subseteq \bar{t}$, let $x^A \in \{0, 1\}^n$ be an assignment such that $x_i^A = \mathbf{1}_{i \in A}, \forall i \neq t$ and $x_t^A = 0$. We define a set function $f_t$ for each terminal node $t \in V$ as follows:

$$f_t(A) = Q_t(X_t = x_t^A) + \sum_{j \in \pi_G(t)} Q_{tj}(X_t = x_t^A, X_j = x_j^A), \quad \forall A \subseteq \bar{t} \tag{3}$$

Note that Equation (3) precisely computes $\mathcal{P}(X_t = 1 | X_{\pi_G(t)} = x_{\pi_G(t)}^A)$ and $f_t(A) = f_t(A \cap \pi_t)$. Next, we prove that the Fourier support of $f_t$ only contains singleton sets of parents of node $t$.

**Theorem 1.** *If nodes of a Bayesian network $\mathcal{B}(G, \mathcal{P})$ have rank 2 with respect to their parents $\pi_G(.)$ and $\mathcal{P}$, then the Fourier coefficient $\hat{f}_t(B)$ for function $f_t$ defined by equation (3) for any terminal node $t$ and a set $B \in 2^{\bar{t}}$ is given by:*

$$\hat{f}_t(B) = \begin{cases} Q_t(X_t = 1) + \frac{1}{2} \sum_{j \in \pi_G(t)} \left( Q_{tj}(X_t = 1, X_j = 0) + Q_{tj}(X_t = 1, X_j = 1) \right), & B = \phi \\ \frac{1}{2} \left( Q_{tj}(X_t = 1, X_j = 0) - Q_{tj}(X_t = 1, X_j = 1) \right), & B = \{j\}, \forall j \in \pi_G(t) \\ 0, & Otherwise \end{cases}$$

$$\tag{4}$$

(See Appendix A for detailed proofs.)

**Analyzing Non-Terminal Nodes.** A similar analysis can be done for non-terminal nodes. However, for a non-terminal node $i$ we can not compute $\mathcal{P}(X_i|X_{\pi_G(i)})$ without explicitly knowing the parents of node $i$. We will rather focus on computing $\mathcal{P}(X_i|X_{\mathrm{MB}_G(i)})$ for non-terminal nodes which equals to computing $\mathcal{P}(X_i|X_{\bar{i}})$ which can be done from data. Similar to the previous case, we define a set function $g_i$ for each non-terminal node $i \in V$ as follows:

$$g_i(A) = Q_i(X_i = x_i^A) + \sum_{j \in \pi_G(i)} Q_{ij}(X_i = x_i^A, X_j = x_j^A), \quad \forall A \subseteq \bar{i} \tag{5}$$

We can define a corresponding set function $f_i$ which computes $\mathcal{P}(X_i|X_{\mathrm{MB}_G(i)})$ for non-terminal nodes. We do it in the following way:

$$
\begin{aligned}
f_i(A) &= \mathcal{P}(X_i = x_i^A | X_{\mathrm{MB}_G(i)} = x_{\mathrm{MB}_G(i)}) \\
&= \frac{\mathcal{P}(X_i = x_i^A | X_{\pi_G(i)} = x_{\pi_G(i)}^A) \prod_{k \in \mathrm{child}_G(i)} \mathcal{P}(X_k = x_k^A | X_{\pi_G(k)} = x_{\pi_G(k)}^A)}{\sum_{X_i} \mathcal{P}(X_i = x_i^A | X_{\pi_G(i)} = x_{\pi_G(i)}^A) \prod_{k \in \mathrm{child}_G(i)} \mathcal{P}(X_k = x_k^A | X_{\pi_G(k)} = x_{\pi_G(k)^A})}
\end{aligned} \tag{6}
$$

where $\mathrm{child}_G(i)$ is the set of children of node $i$ in DAG $G$. We can again compute the Fourier support for $f_i$ for each non-terminal node.

**Theorem 2.** *If nodes of a Bayesian network $\mathcal{B}(G, \mathcal{P})$ have rank 2 with respect to their parents $\pi_G(.)$ and $\mathcal{P}$, then the Fourier coefficient $\hat{f}_i(B)$ for function $f_i$ defined by equation* (6) *for any non-terminal node $i$ and a set $B \in 2^{\bar{i}}$ is given by:*

$$
\hat{f}_i(B) = \begin{cases} 0, & |B \setminus \mathrm{MB}_G(i)| \geqslant 1 \\ \frac{1}{2^{n-1}} \sum_{A \in 2^{V-i}} \frac{g_i(A) \prod_{k \in \mathrm{child}_G(i)} g_k(A)}{g_i(A) \prod_{k \in \mathrm{child}_G(i)} g_k(A) + g_i(A \cup \{i\}) \prod_{k \in \mathrm{child}_G(i)} g_k(A \cup \{i\})} \psi_B(A), & \textit{otherwise} \end{cases} \tag{7}
$$

## 3.3 Algorithm

Our algorithm works on the principle that the terminal nodes are rank 2 with respect to their Markov Blanket and $\mathcal{P}$, while non-terminal nodes are not. This is true if for every non-terminal node there exists a $B \in 2^V$ such that $|B \setminus \mathrm{MB}_G(i)| = 0$ and $\hat{f}_i(B)$ is nonzero. This is formalized in what follows.

**Assumption 4** (Non-terminal nodes are not rank 2). *There exists a $B \in 2^V$ for each non-terminal node $i$, such that $|B| = 2$, $|B \setminus \mathrm{MB}_G(i)| = 0$ and $\hat{f}_i(B)$ as defined by Equation* (7) *is non-zero.*

This distinction helps us to differentiate between terminal and non-terminal nodes. Note that the set function $f_i$ is uniquely determined by its Fourier coefficients. Moreover, the Fourier support for function $f_i$ is sparse. For terminal nodes, $\hat{f}_i(B)$ is non-zero only for the empty set or the singleton nodes, while for the non-terminal nodes, $\hat{f}_i(B)$ is non-zero for $B \subseteq \mathrm{MB}_G(i)$. Thus, recovering Fourier coefficients from the measurements of $f_i$ can be treated as recovering a sparse vector in $\mathbb{R}^{2^{\bar{i}}}$. However, $|2^{\bar{i}}|$ could be quite large. We avoid this problem by substituting $f_i$ by another function $g_i \in \mathcal{G}_2$ where $\mathcal{G}_k = \{g_i \mid \forall B \in \mathrm{support}(g_i), |B| \leqslant k\}$. Note that,

$$f_i(A_j) = \sum_{\substack{|B_k|=1 \\ B_k \in 2^{\bar{i}}}} \hat{f}_i(B_k)\psi_{B_k}(A_j) + \sum_{\substack{|B_k|=2 \\ B_k \in 2^{\bar{i}}}} \hat{f}_i(B_k)\psi_{B_k}(A_j) + \sum_{\substack{|B_k|\geqslant 3 \\ B_k \in 2^{\bar{i}}}} \hat{f}_i(B_k)\psi_{B_k}(A_j) \tag{8}$$

and $\forall g_i \in \mathcal{G}_2$,

$$g_i(A_j) = \sum_{\substack{|B_k|=1 \\ B_k \in 2^{\bar{i}}}} \hat{f}_i(B_k)\psi_{B_k}(A_j) + \sum_{\substack{|B_k|=2 \\ B_k \in 2^{\bar{i}}}} \hat{f}_i(B_k)\psi_{B_k}(A_j) \tag{9}$$

It follows that for a terminal node $i$, $g_i = f_i$ as for terminal nodes $f_i \in \mathcal{G}_1$. For non-terminal nodes, using results from Theorem 2, if $B \subseteq \mathrm{MB}_G(i)$ then $\hat{f}_i(B) \neq 0$ and therefore $g_i \notin \mathcal{G}_1$. Now, let $\mathcal{A}_i$

be a collection of $m_i$ sets $A_j \in 2^{V-i}$ chosen uniformly at random. We measure $g_i(A_j)$ for each $A_j \in \mathcal{A}_i$ and then using equation (2) we can write:

$$g_i(A_j) = \sum_{B_k \in 2^{\bar{i}}, |B_k| \leqslant 2} (-1)^{|A_j \cap B_k|} \hat{f}_i(B_k), \forall A_j \in \mathcal{A}_i \tag{10}$$

Let $\boldsymbol{g_i} \in \mathbb{R}^{m_i}$ be a vector whose $j^{\text{th}}$ row is $g_i(A_j)$ and $\hat{\boldsymbol{g_i}} \in \mathbb{R}^{n+\binom{n-1}{2}}$ be a vector with elements of form $\hat{f}_i(B_k) \forall B_k \in \rho_i$ where

$$\rho_i = \{B_k \mid B_k \in 2^{\bar{i}}, |B_k| \leqslant 2\} \tag{11}$$

is a set which contains support$(\hat{f}_i)$. Then,

$$\boldsymbol{g_i} = \mathcal{M}_i \hat{\boldsymbol{g_i}} \quad \text{where, } \mathcal{M}_i \in \{-1,1\}^{m_i \times n} \text{ such that } \mathcal{M}^i_{jk} = (-1)^{|A_j \cap B_k|} . \tag{12}$$

Also note that for terminal nodes $\hat{\boldsymbol{g_i}}$ is sparse with $|\pi_G(i)| + 1$ non-zero elements for terminal nodes and at max $\binom{k}{2} + k + 1$ non-zero elements for non-terminal nodes where $k = |\text{MB}_G(i)|$. Equation (12) can be solved by any standard compressed sensing techniques to recover the parents of the terminal nodes. Using this formulation and the fact that terminal nodes have non-zero Fourier coefficients on empty or singleton sets, we can provide an algorithm to identify the terminal nodes and their corresponding parents. We can use this algorithm repeatedly to identify the complete structure of the Bayesian network until the last two nodes where we can not apply our algorithm. Algorithm 1 identifies the parents for each node and consequently the directed structure of the Bayesian network.

**Algorithm 1:** getParents($V$)

**Input** : Nodes $V = \{1, 2, \ldots, n\}$
**Output** : Recovered parent set $\hat{\pi} : V \to 2^V$
$S \leftarrow V$ ;
**while** $|S| \geqslant 3$ **do**
  $T, \hat{\pi} = \text{getTerminalNodes}(S)$ ;
  $S \leftarrow S \backslash T$ ;
**end**
**for** $i \in S$ **do**
  $\hat{\pi}(i) = \phi$;
**end**

**Algorithm 2:** getTerminalNodes($S$)

**Input** : Nodes $S \subseteq \{1, 2, \ldots, n\}$
**Output** : Set of terminal nodes $T$ and their parents
      $\hat{\pi} : T \to 2^S$
$T \leftarrow \phi, \hat{\pi}(i) \leftarrow \phi \; \forall i \in S$ ;
**for** *node* $i \in S, j \in \{1, \ldots, m_i\}$ **do**
  Choose $A_j \in 2^{S \backslash \{i\}}$ uniformly at random ;
  Compute
    $f_i(A_j) = \mathcal{P}(X_i = 0 | X_{S \backslash \{i\}} = x^{A_j}_{S \backslash \{i\}})$ ;
  Compute $\mathcal{M}^i_{jk}$ for $B_k \in \rho_i$ (Eq (11) (12)) ;
  Solve for $\boldsymbol{\beta_i}$ using compressed sensing (Eq
   (13)) ;
  **if** $\boldsymbol{\beta}_i(B) = 0 \text{ for all } |B| > 1$ **then**
    $T \leftarrow T \cup \{i\}$ ;
    $\hat{\pi}(i) \leftarrow \cup_{B:\boldsymbol{\beta}_i(B) \neq 0} B$ ;
  **end**
**end**

## 4 Analysis in Finite Sample Regime

So far our results have been in the population setting where we assumed that we had access to the true conditional probabilities. However, generally this is not the case and we have to work with a finite number of samples from the *black-box*. In this section, we provide theoretical results for different finite sample regimes.

### 4.1 Without access to any observational data

In this setting, we assume that we only have access to a black-box which outputs observational data for our conditional probability queries. One selection of nodes consists of fixing $X_{\bar{i}}$ and then measuring $X_i$ for each node $i$. We need only 1 selection for each node. Thus the total number of selections for all the nodes is $n$. One query amounts to fixing $X_{\bar{i}}$ to a particular $x_{\bar{i}}$. Note that while $2^{n-1}$ such queries are possible for each selection on each node, we only conduct $m_i$ queries for each node $i$.

**Number of Queries.** We measure $g_i(A_j)$ by querying for $f_i(A_j)$. Let $|f_i(A_j) - g_i(A_j)| \leqslant \epsilon_j, \forall A_j \in \mathcal{A}_i$ for some $\epsilon_j > 0$. Once we have the noisy measurements of $g_i(A_j)$, we can get a good approximation of $\hat{\boldsymbol{g}}_{\boldsymbol{i}}$ by solving the following optimization problem for each node $i$.

$$\boldsymbol{\beta}_i = \min_{\hat{\boldsymbol{g}}_{\boldsymbol{i}} \in \mathbb{R}^{|\rho_i|}} \|\hat{\boldsymbol{g}}_{\boldsymbol{i}}\|_1 \quad s.t. \|\mathcal{M}_i \hat{\boldsymbol{g}}_{\boldsymbol{i}} - \boldsymbol{f_i}\|_2 \leqslant \epsilon \quad \text{where } \epsilon = \sqrt{\sum_{A_j \in \mathcal{A}_j} \epsilon_j^2}. \tag{13}$$

**Theorem 3.** *Suppose $\hat{\boldsymbol{g}}_{\boldsymbol{i}}$ is constructed by computing $\hat{g}_i(B_k)$ using $B_k$ from a fixed collection $\rho_i$ as defined in Equation* (11). *Furthermore, suppose $\boldsymbol{g_i}$ is computed by selecting $m_i$ sets $A_j$ uniformly at random from $2^{\bar{i}}$. We define the matrix $\mathcal{M}_i$ as in equation* (12). *Then there exist universal constants $C_1, C_2 > 0$ such that if, $m_i \geqslant \max(C_1|support(\hat{g}_i)| \log^4(n + \binom{n-1}{2}), C_2|support(\hat{g}_i)| \log \frac{1}{\delta})$ and $\boldsymbol{\beta_i}$ is solved using equation* (13). *Then with probability at least $1 - \delta$, we have $\|\boldsymbol{\beta_i} - \hat{\boldsymbol{g}}_{\boldsymbol{i}}\|_2 \leqslant C_3 \frac{\epsilon}{\sqrt{m_i}}$ for some universal constant $C_3 > 0$. If the minimum non-zero element of $|\hat{\boldsymbol{g}}_{\boldsymbol{i}}|$ is greater than $2C_3 \frac{\epsilon}{\sqrt{m_i}}$ then $\boldsymbol{\beta_i}$ recovers $\hat{\boldsymbol{g}}_{\boldsymbol{i}}$ up to the signs. Furthermore, if Assumption 4 is satisfied then $|\boldsymbol{\beta_i}(B)| \leqslant C_3 \frac{\epsilon}{\sqrt{m_i}}, \forall B \in \rho_i, |B| = 2$ if and only if $i$ is a terminal node and $\hat{\pi}(i) = \{B \mid |B| = 1, |\boldsymbol{\beta_i}(B)| > C_3 \frac{\epsilon}{\sqrt{m_i}}\}$ correctly recovers the parents of the terminal node $i$, i.e., $\hat{\pi}(i) = \pi_G(i)$. Applying this recursively shows the correctness of Algorithm 1.*

The sparsity for each node would be less than or equal to $\binom{k}{2} + k + 1$. Thus the number of queries needed for each node (using arguments from Theorem 3) would be of order $\mathcal{O}(\max(k^2 \log^4 n, k^2 \log \frac{1}{\delta}))$. At the first iteration, we query all the nodes. From the next iteration onwards, we query for only the nodes which had terminal nodes as their children,i.e., for a maximum of $k$ nodes. Thus the total number of queries needed would be $\mathcal{O}(\max(nk^3 \log^4 n, nk^3 \log \frac{1}{\delta}))$. We can recover parents for terminal nodes using Theorem 3.

**Sample and Time Complexity.** The sample complexity is $\mathcal{O}(\max(\frac{nk^3 \log^4 n}{\epsilon^2}(\log k + \log \log n), \frac{nk^3}{\epsilon^2} \log \frac{1}{\delta}(\log k + \log \log n))$ and the time complexity is $\mathcal{O}(n^4 k \sqrt{n} \log n)$ (See Appendix B for details).

## 4.2 With Access to Some Observational Data

In this setting, we have access to some observational data as well. We can use the observational data to figure out the Markov blanket of each node which helps us reduce number of selected variables in the conditional probability queries. Once we have the Markov blanket, we only select the nodes in $\text{MB}_G(i)$ for each query. We need only 1 selection for each node. Thus the total number of selections for all nodes does not exceed $n$.

**Using Observational Data.** Recall that $\mathcal{P}$ is the true joint distribution over the nodes of a Bayesian network $\mathcal{B}(G, \mathcal{P})$. We define a collection of distributions $\mathbb{P}$ over the nodes of the Bayesian network as: $\mathbb{P} = \{P \text{ is faithful to } G. |P(X_i|X_l) = \mathcal{P}(X_i|X_l), \forall i, l \in \{1, \dots, n\}\}$

**Computing the Markov Blanket from Observational Data.** Consider a probability distribution $\hat{P} \in \mathbb{P}$ on the nodes of the Bayesian network such that each node $i$ is rank 2 with respect to $\text{MB}_G(i)$ and $\hat{P}$. This allows us to recover the Markov blanket of the node using the observational data.

**Theorem 4.** *If there exists a probability distribution $\hat{P} \in \mathbb{P}$ such that each node $i$ is rank 2 with respect to $\text{MB}_G(i)$ and $\hat{P}$, then the Markov blanket of a node $i$ can be recovered by solving the following system of equations:*

$$\mathcal{P}(X_i = 0, X_l = 0) = \tilde{Q}_i(X_i = 0)\mathcal{P}(X_l = 0) + \sum_{\substack{j \in -i \\ j \neq l}} \tilde{Q}_{ij}(X_i = 0, X_j = 0)\mathcal{P}(X_j = 0, X_l = 0)$$

$$+ \tilde{Q}_{il}(X_i = 0, X_l = 0)\mathcal{P}(X_l = 0), \ \forall \ l = \{1, \dots, n\}, l \neq i$$

$$\mathcal{P}(X_i = 0) = \tilde{Q}_i(X_i = 0) + \sum_{\substack{j \in -i \\ j \neq l}} \tilde{Q}_{ij}(X_i = 0, X_j = 0)\mathcal{P}(X_j = 0)$$

*which can be written in a more compact form:*

$$\bar{\boldsymbol{y}} = \overline{\boldsymbol{A}}\boldsymbol{q} \tag{14}$$

*where $\bar{\boldsymbol{y}} \in \mathbb{R}^n$ and $\overline{\boldsymbol{A}} \in \mathbb{R}^{n \times n}$ and $\boldsymbol{q} \in \mathbb{R}^n$. The entries of $\bar{\boldsymbol{y}}$ are indexed by $l = \{1 \dots n\}$ such that $\bar{\boldsymbol{y}}_l = \mathcal{P}(X_i = 0, X_l = 0)$ when $l \neq i$ and $\bar{\boldsymbol{y}}_l = \mathcal{P}(X_i = 0)$ when $l = i$. The entries of $\overline{\boldsymbol{A}}$ are indexed by $l, j \in \{1, \dots, n\}$, where $\overline{\boldsymbol{A}}_{lj} = \mathcal{P}(X_l = 0, X_j = 0)$ for $l \neq i, j \neq i, j \neq l$ and , $\overline{\boldsymbol{A}}_{lj} = \mathcal{P}(X_l = 0)$ when $l = j, l \neq i$, $\overline{\boldsymbol{A}}_{lj} = \mathcal{P}(X_l = 0)$ for $l \neq i, j = i$, $\overline{\boldsymbol{A}}_{lj} = \mathcal{P}(X_j = 0)$ for $l = i, j \neq i$ and $\overline{\boldsymbol{A}}_{lj} = 1$ for $l = i, j = i$. The entries of $\boldsymbol{q}$ are indexed by $j \in \{1, \dots, n\}$ such that $\boldsymbol{q}_j = \tilde{Q}_{ij}(X_i = 0, X_j = 0)$ for $j \neq i$ and $\boldsymbol{q}_j = \tilde{Q}_i(X_i = 0)$ for $j = i$.*

For terminal nodes, existence of $\hat{P} \in \mathbb{P}$ as $\mathcal{P} \in \mathbb{P}$ is guaranteed. To ensure that $\hat{P} \in \mathbb{P}$ also exists for non-terminal nodes, we make the following assumption:

**Assumption 5.** *The population matrix $\overline{\boldsymbol{A}} \in \mathbb{R}^{n \times n}$ as defined in equation (14) is positive definite.*

This assumption is not strong. We can, in fact, show that $\overline{\mathbf{A}}$ is a positive semidefinite matrix.

**Lemma 1.** *The population matrix $\overline{\boldsymbol{A}}$ as defined in equation (14) is a positive semidefinite matrix.*

We can solve Equation (14) to get $\tilde{Q}_i$ and $\tilde{Q}_{ij}$. The Markov blanket of node $i$ is computed by $\mathrm{MB}_G(i) = \{j \mid \tilde{Q}_{ij} \neq 0\}$. To this end, we prove that:

**Lemma 2.** *If $\tilde{Q}_{ij}(\cdot, \cdot), \forall j \in \{1, \dots, n\}, j \neq i$ is computed by solving system of linear equations (14) and $\hat{P} \in \mathbb{P}$ is faithful to $G$ then $\tilde{Q}_{ij}(\cdot, \cdot) \neq 0, \forall j \in \{1, \dots, n\}, j \neq i$ if and only if $j \in \mathrm{MB}_G(i)$.*

Once we know the Markov blanket for each node $i$, the queries in Algorithm 2 can be changed from $\boldsymbol{f_i}(A_j) = \mathcal{P}(X_i = 0 | X_{S \setminus \{i\}} = x^{A_j}_{S \setminus \{i\}})$ to $\boldsymbol{f_i}(A_j) = \mathcal{P}(X_i = 0 | X_{S \cap \mathrm{MB}_i} = x^{A_j}_{S \cap \mathrm{MB}_i})$ which helps in reducing the sample and time complexity.

**Number of Queries.** Again, let $|\mathrm{MB}_G(i)| \leqslant k, \forall i \in \{1, \dots, n\}$. The sparsity for each node would be less than or equal to $\binom{k}{2} + k + 1$. Thus number of queries needed for each node (using arguments from Theorem 3) would be of order $\mathcal{O}(\max(k^2 \log^4 k, k^2 \log \frac{1}{\delta}))$. As before, these queries are repeated $nk$ times. Thus the total number of queries needed would be $\mathcal{O}(\max(nk^3 \log^4 k, nk^3 \log \frac{1}{\delta}))$.

**Sample and Time Complexity.** We use the following lemma to get the sample complexity for the observational data.

**Lemma 3.** *$N = \mathcal{O}(\frac{\log n}{\epsilon^2})$ i.i.d observations are sufficient to measure elements of $\overline{\boldsymbol{A}}$ and $\bar{\boldsymbol{y}}$, $\epsilon$ close to their true value. That is $|\overline{\boldsymbol{A}} - \hat{\boldsymbol{A}}| \leqslant \epsilon$ and $|\bar{\boldsymbol{y}} - \hat{\boldsymbol{y}}| \leqslant \epsilon$, for some $\epsilon > 0$ with probability at least $1 - 2\exp(\log(\binom{n}{2} + 3n) - \frac{N\epsilon^2}{2})$ for some $\epsilon > 0$ where $\hat{\boldsymbol{A}}$ and $\hat{\boldsymbol{y}}$ are the empirical measurements of $\overline{\boldsymbol{A}}$ and $\bar{\boldsymbol{y}}$ respectively and $|\cdot - \cdot|$ denotes componentwise comparison for matrices.*

At this point, it remains to be shown that we can still recover the Markov blanket for the nodes using the noisy measurements of unary and pairwise marginals. Below, we prove that this is true as long as $\overline{\mathbf{A}}$ is well conditioned.

**Lemma 4.** *Let $\hat{\boldsymbol{A}}$ and $\hat{\boldsymbol{y}}$ be the empirical measurements of $\overline{\boldsymbol{A}}$ and $\bar{\boldsymbol{y}}$ as defined in equation (14) respectively such that $|\hat{\boldsymbol{A}} - \overline{\boldsymbol{A}}| \leqslant \epsilon$ and $|\hat{\boldsymbol{y}} - \bar{\boldsymbol{y}}| \leqslant \epsilon$ for some $\epsilon > 0$, where $|\cdot - \cdot|$ denotes componentwise comparison for matrices. Let $\hat{\boldsymbol{q}}$ be the solution to the system of linear equations given by $\hat{\boldsymbol{y}} = \hat{\boldsymbol{A}}\hat{\boldsymbol{q}}$ and $\eta \kappa_\infty(\overline{\boldsymbol{A}}) \leqslant 1$, then $\hat{\boldsymbol{q}}$ recovers $\boldsymbol{q}$ up to signs as long as $N = \mathcal{O}(n)$ i.i.d. measurements are used to measure $\hat{\boldsymbol{A}}$ and $\frac{\max_i |\boldsymbol{q}_i|}{\min_i |\boldsymbol{q}_i|} \leqslant \frac{1 - \eta \kappa_\infty(\overline{\boldsymbol{A}})}{4\eta \kappa_\infty(\overline{\boldsymbol{A}})}$, where $\kappa_\infty(\overline{\boldsymbol{A}}) \triangleq \|\overline{\boldsymbol{A}}\|_\infty \|\overline{\boldsymbol{A}}^{-1}\|_\infty$ is the condition number of $\overline{\boldsymbol{A}}$ and $\eta = \max(\frac{n\epsilon}{\sum_{j=1}^{n-1} \mathcal{P}(X_j = 0) + 1}, \frac{\epsilon}{\mathcal{P}(X_n = 0)})$.*

The time complexity of computing the Markov Blanket is $\mathcal{O}(n^4)$. The sample complexity for the black-box queries is $\mathcal{O}(\max(\frac{nk^3 \log^5 k}{\epsilon^2}, \frac{nk^3}{\epsilon^2} \log \frac{1}{\delta} \log k))$ and the time complexity is $\mathcal{O}(nk^4 \sqrt{k} \log k)$ (See Appendix C for details).

For synthetic experiments validating our theory, please See Appendix D.

**Concluding Remarks.** In this paper, we provide a novel method with theoretical guarantees to recover directed structure of a Bayesian network using black-box queries. We further improve our results when we have access to some observational data. We developed a theory for rank 2 CPTs which can easily be extended to a more general rank $k$ CPTs. It would be interesting to see if we can provide similar results for a Bayesian network with low rank CPTs using pure observational or interventional data.

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
