[Supplementary Material]

# Appendix

## A  Detailed Proofs of Theorem and Lemmas

### A.1  Proof of Theorem 1

**Theorem 1**  *If nodes of a Bayesian network $\mathcal{B}(G, \mathcal{P})$ have rank 2 with respect to their parents $\pi_G(.)$ and $\mathcal{P}$, then the Fourier coefficient $\hat{f}_t(B)$ for function $f_t$ defined by equation (3) for any terminal node $t$ and a set $B \in 2^{\bar{t}}$ is given by:*

$$\hat{f}_t(B) = \begin{cases} Q_t(X_t = 1) + \frac{1}{2}\sum_{j \in \pi_G(t)}\left(Q_{tj}(X_t = 1, X_j = 0) + Q_{tj}(X_t = 1, X_j = 1)\right), & B = \phi \\ \frac{1}{2}\left(Q_{tj}(X_t = 1, X_j = 0) - Q_{tj}(X_t = 1, X_j = 1)\right), & B = \{j\}, \forall j \in \pi_G(t) \\ 0, & Otherwise \end{cases}$$

$$(15)$$

*Proof.* The Fourier transformation coefficients $\hat{f}_t$ can be calculated using the following formula:

$$\hat{f}_t(B) = 2^{-n+1}\sum_{A \in 2^{\bar{t}}} f_t(A)(-1)^{|A \cap B|} \qquad (16)$$

We prove our claim by computing $\hat{f}_t(B)$ explicitly for various setting of $B \in 2^{\bar{t}}$.

**Case 1.** $B = \phi$.

$$\begin{aligned}
\hat{f}_t(B) &= 2^{-n+1}\sum_{A \in 2^{\bar{t}}} f_t(A)(-1)^{|A \cap B|} \\
&= 2^{-n+1}\sum_{A \in 2^{\bar{t}}} f_t(A), \quad |A \cap B| = 0 \\
&= 2^{-n+1}\sum_{A \in 2^{\bar{t}}}[Q_t(X_t = 1) + \sum_{j \in \pi_G(t)} Q_{tj}(X_t = 1, X_j = x_j^A)] \\
&= 2^{-n+1}2^{n-1}Q_t(X_t = 1) + 2^{-n+1}\sum_{A \in 2^{\bar{t}}}\sum_{j \in \pi_G(t)} Q_{tj}(X_t = 1, X_j = x_j^A) \\
&= Q_t(X_t = 1) + \frac{1}{2}\sum_{j \in \pi_G(t)}[Q_{tj}(X_t = 1, X_j = 0) + Q_{tj}(X_t = 1, X_j = 1)]
\end{aligned}$$

**Case 2.** $B = \{l\}, l \in \pi_G(t)$.

$$\begin{aligned}
\hat{f}_t(B) &= 2^{-n+1}\sum_{A \in 2^{\bar{t}}} f_t(A)(-1)^{|A \cap B|} \\
&= 2^{-n+1}[-\sum_{A \in 2^{\bar{t}}, l \in A} f_t(A) + \sum_{A \in 2^{\bar{t}}, l \notin A} f_t(A)] \\
&= 2^{-n+1}[-\sum_{A \in 2^{\bar{t}}, l \in A}[Q_t(X_t = 1) + Q_{tl}(X_t = 1, X_l = 1) + \sum_{j \in \pi_G(t)-l} Q_{tj}(X_t = 1, X_j = x_j^A)] \\
&\quad + \sum_{A \in 2^{\bar{t}}, l \notin A}[Q_t(X_t = 1) + Q_{tl}(X_t = 1, X_l = 0) + \sum_{j \in \pi_G(t)-l} Q_{tj}(X_t = 1, X_j = x_j^A)]] \\
&= \frac{1}{2}[Q_{tl}(X_t = 1, X_l = 0) - Q_{tl}(X_t = 1, X_l = 1)]
\end{aligned}$$

**Case 3.** $B \subseteq \pi_G(t), |B| > 1.$

$$\hat{f}_t(B) = 2^{-n+1} \sum_{A \in 2^{\bar{t}}} f_t(A)(-1)^{|A \cap B|}$$

$$= 2^{-n+1} \sum_{A \in 2^{\bar{t}}} [[Q_t(X_t = 1) + \sum_{j \in \pi_G(t)} Q_{tj}(X_t = 1, X_j = x_j^A)](-1)^{|A \cap B|}]$$

Take an $l \in B \implies l \in \pi_G(t)$

$$= 2^{-n+1}[\sum_{A \in 2^{\bar{t}}, l \notin A} [Q_t(X_t = 1) + Q_{tl}(X_t = 1, X_l = 0)$$

$$+ \sum_{j \in \pi_G(t)-l} Q_{tj}(X_t = 1, X_j = x_j^A)](-1)^{|A \cap B-l|} + \sum_{A \in 2^{\bar{t}}, l \in A} [Q_t(X_t = 1) + Q_{tl}(X_t = 1, X_l = 1)$$

$$+ \sum_{j \in \pi_G(t)-l} Q_{tj}(X_t = 1, X_j = x_j^A)](-1)^{1+|A \cap B-l|}]$$

$$= 2^{-n+1}[\sum_{A \in 2^{\bar{t} \backslash \{l\}}} [Q_{tl}(X_t = 1, X_l = 0) - Q_{tl}(X_t = 1, X_l = 1)](-1)^{|A \cap B|}]$$

Take $k \in B$

$$= 2^{-n+1}[\sum_{A \in 2^{\bar{t} \backslash \{l\}}, k \in A} [Q_{tl}(X_t = 1, X_l = 0) - Q_{tl}(X_t = 1, X_l = 1)](-1)^{1+|A \cap B-k|}]$$

$$+ 2^{-n+1}[\sum_{A \in 2^{\bar{t} \backslash \{l\}}, k \notin A} [Q_{tl}(X_t = 1, X_l = 0) - Q_{tl}(X_t = 1, X_l = 1)](-1)^{|A \cap B-k|}]$$

$$= 0$$

**Case 4.** $|B \cap \bar{t} - \pi_G(t)| \geqslant 1$

$$\hat{f}_t(B) = 2^{-n+1} \sum_{A \in 2^{\bar{t}}} f_t(A)(-1)^{|A \cap B|}$$

$$= 2^{-n+1} \sum_{A \in 2^{\bar{t}}} f_t(A \cap \pi_G(t))(-1)^{|A \cap B|}$$

Take $l \in B$ and $l \notin \pi_G(t)$

$$= 2^{-n+1}[\sum_{A \in 2^{\bar{t}}, l \notin A} f_t(A \cap \pi_G(t))(-1)^{|A \cap B-l|} + \sum_{A \in 2^{\bar{t}}, l \in A} f_t(A \cap \pi_G(t))(-1)^{1+|A \cap B-l|}]$$

$$= 0$$

This proves our claim. $\qquad\square$

### A.2 Proof of Theorem 2

**Theorem 2** *If nodes of a Bayesian network $\mathcal{B}(G, \mathcal{P})$ have rank 2 with respect to their parents $\pi_G(.)$ and $\mathcal{P}$, then the Fourier coefficient $\hat{f}_i(B)$ for function $f_i$ defined by equation (6) for any non-terminal node $i$ and a set $B \in 2^{\bar{i}}$ is given by:*

$$\hat{f}_i(B) = \begin{cases} 0, & |B \backslash \mathrm{MB}_G(i)| \geqslant 1 \\ \frac{1}{2^{n-1}} \sum_{A \in 2^{V-i}} \frac{g_i(A)\prod_{k \in \mathrm{child}_G(i)} g_k(A)}{g_i(A)\prod_{k \in \mathrm{child}_G(i)} g_k(A) + g_i(A \cup \{i\})\prod_{k \in \mathrm{child}_G(i)} g_k(A \cup \{i\})} \psi_B(A), & otherwise \end{cases}$$

$$(17)$$

*Proof.* Note that for the case $|B - \mathrm{MB}_G(i)| = 0$, we simply replace terms in Equation (6) with appropriate set functions. It can be simplified for various cases but we chose not to do it for clarity of representation. For the second case when $|B - \mathrm{MB}_G(i)| \geqslant 1$, $\exists s$ such that $s \in B$ and $s \notin \mathrm{MB}_G(i)$.

Note that $f_i(A) = f_i(A \cap \mathrm{MB}_G(i))$. Take $A = A' \cup \{s\}$ and $s \notin A'$.

$$\hat{f}_i(B) = 2^{-n+1} \sum_{A \in 2^{V-i}} f_i(A)(-1)^{|A \cap B|}$$

$$= 2^{-n+1}\left( \sum_{A' \in 2^{V-\{i,s\}}} f_i(A')(-1)^{|A' \cap B|+1} + \sum_{A' \in 2^{V-\{i,s\}}} f_i(A')(-1)^{|A' \cap B|} \right) \quad = 0$$

$\square$

### A.3 Proof of Theorem 3

**Theorem 3** *Suppose $\hat{\boldsymbol{g}}_{\boldsymbol{i}}$ is constructed by computing $\hat{g}_i(B_k)$ using $B_k$ from a fixed collection $\rho_i$ as defined in Equation (11). Furthermore, suppose $\boldsymbol{g}_{\boldsymbol{i}}$ is computed by selecting $m_i$ sets $A_j$ uniformly at random from $2^{\bar{i}}$. We define the matrix $\mathcal{M}_i$ as in equation (12). Then there exist universal constants $C_1, C_2 > 0$ such that if, $m_i \geqslant \max(C_1|\mathrm{support}(\hat{g}_i)| \log^4(n + \binom{n-1}{2}), C_2|\mathrm{support}(\hat{g}_i)| \log \frac{1}{\delta})$ and $\boldsymbol{\beta}_{\boldsymbol{i}}$ is solved using equation (13). Then with probability at least $1 - \delta$, we have $\|\boldsymbol{\beta}_{\boldsymbol{i}} - \hat{\boldsymbol{g}}_{\boldsymbol{i}}\|_2 \leqslant C_3 \frac{\epsilon}{\sqrt{m_i}}$ for some universal constant $C_3 > 0$. If the minimum non-zero element of $|\hat{\boldsymbol{g}}_{\boldsymbol{i}}|$ is greater than $2C_3 \frac{\epsilon}{\sqrt{m_i}}$ then $\boldsymbol{\beta}_i$ recovers $\hat{\boldsymbol{g}}_{\boldsymbol{i}}$ up to the signs. Furthermore, if Assumption 4 is satisfied then $|\boldsymbol{\beta}_i(B)| \leqslant C_3 \frac{\epsilon}{\sqrt{m_i}}, \forall B \in \rho_i, |B| = 2$ if and only if $i$ is a terminal node and $\hat{\pi}(i) = \{B \mid |B| = 1, |\boldsymbol{\beta}_i(B)| > C_3 \frac{\epsilon}{\sqrt{m_i}}\}$ correctly recovers the parents of the terminal node $i$, i.e., $\hat{\pi}(i) = \pi_G(i)$. Applying this recursively shows the correctness of Algorithm 1.*

*Proof.* First note that the rows of $\mathcal{M}_i$ are sampled uniformly at random from an orthonormal matrix with bounded entries. Rauhut (2010) have proved that Restricted Isometry Property (RIP) holds for such matrices with high probability. Thus, we can invoke Theorem 1 from Stobbe and Krause (2012) which in turn follows the proof of Theorem 4.4 from Rauhut (2010) to get the result that $\|\boldsymbol{\beta}_{\boldsymbol{i}} - \hat{\boldsymbol{g}}_{\boldsymbol{i}}\|_2 \leqslant C_3 \frac{\epsilon}{\sqrt{m_i}}$.

Furthermore, $\|\boldsymbol{\beta}_i - \hat{\boldsymbol{g}}_{\boldsymbol{i}}\|_\infty \leqslant \|\boldsymbol{\beta}_i - \hat{\boldsymbol{g}}_{\boldsymbol{i}}\|_2 \leqslant C_3 \frac{\epsilon}{\sqrt{m_i}}$. Thus if the minimum non-zero element of $|\hat{\boldsymbol{g}}_{\boldsymbol{i}}|$ is greater than $2C_3 \frac{\epsilon}{\sqrt{m_i}}$ then $\boldsymbol{\beta}_i$ recovers $\hat{\boldsymbol{g}}_{\boldsymbol{i}}$ up to the signs.

Adding to the above, the results from Theorem 1 and Assumption 4 ensure that $|\boldsymbol{\beta}_i(B)| \leqslant C_3 \frac{\epsilon}{\sqrt{m_i}}, \forall B \in \rho_i, |B| = 2$ if and only if $i$ is a terminal node and $\hat{\pi}(i) = \pi_G(i)$. $\square$

### A.4 Proof of Theorem 4

**Theorem 4** *If there exists a probability distribution $\hat{P} \in \mathbb{P}$ such that each node $i$ is rank 2 with respect to $\mathrm{MB}_G(i)$ and $\hat{P}$, then the Markov blanket of a node $i$ can be recovered by solving the following system of equations:*

$$\mathcal{P}(X_i = 0, X_l = 0) = \tilde{Q}_i(X_i = 0)\mathcal{P}(X_l = 0) + \sum_{\substack{j \in -i \\ j \neq l}} \tilde{Q}_{ij}(X_i = 0, X_j = 0)\mathcal{P}(X_j = 0, X_l = 0)$$

$$+ \tilde{Q}_{il}(X_i = 0, X_l = 0)\mathcal{P}(X_l = 0), \ \forall \ l = \{1, \dots, n\}, l \neq i$$

$$\mathcal{P}(X_i = 0) = \tilde{Q}_i(X_i = 0) + \sum_{\substack{j \in -i \\ j \neq l}} \tilde{Q}_{ij}(X_i = 0, X_j = 0)\mathcal{P}(X_j = 0)$$

(18)

*which can be written in a more compact form:*

$$\bar{\boldsymbol{y}} = \bar{\boldsymbol{A}}\boldsymbol{q} \tag{19}$$

*where $\bar{\boldsymbol{y}} \in \mathbb{R}^n$ and $\bar{\boldsymbol{A}} \in \mathbb{R}^{n \times n}$ and $\boldsymbol{q} \in \mathbb{R}^n$. The entries of $\bar{\boldsymbol{y}}$ are indexed by $l = \{1 \dots n\}$ such that $\bar{\boldsymbol{y}}_l = \mathcal{P}(X_i = 0, X_l = 0)$ when $l \neq i$ and $\bar{\boldsymbol{y}}_l = \mathcal{P}(X_i = 0)$ when $l = i$. The entries of $\bar{\boldsymbol{A}}$ are indexed by $l, j \in \{1, \dots, n\}$, where $\bar{\boldsymbol{A}}_{lj} = \mathcal{P}(X_l = 0, X_j = 0)$ for $l \neq i, j \neq i, j \neq l$ and, $\bar{\boldsymbol{A}}_{lj} = \mathcal{P}(X_l = 0)$*

*when* $l = j, l \neq i$, $\overline{A}_{lj} = \mathcal{P}(X_l = 0)$ *for* $l \neq i, j = i$, $\overline{A}_{lj} = \mathcal{P}(X_j = 0)$ *for* $l = i, j \neq i$ *and* $\overline{A}_{lj} = 1$ *for* $l = i, j = i$. *The entries of* $\boldsymbol{q}$ *are indexed by* $j \in \{1, \ldots, n\}$ *such that* $\boldsymbol{q}_j = \tilde{Q}_{ij}(X_i = 0, X_j = 0)$ *for* $j \neq i$ *and* $\boldsymbol{q}_j = \tilde{Q}_i(X_i = 0)$ *for* $j = i$.

*Proof.* If there exists a probability distribution $\hat{P} \in \mathbb{P}$ such that each node $i$ is rank 2 with respect to $\mathrm{MB}_G(i)$ and $\hat{P}$, then

$$\hat{P}(X_i = x_i | X_{-i}) = Q_i(X_i = x_i) + \sum_{j \in -i} Q_{ij}(X_i = x_i, X_j) \tag{20}$$

where $Q_{ij}(X_i = x_i, X_j) = 0$ if $j \notin \mathrm{MB}_G(i)$.

For nodes $i, l \in \{1, \ldots, n\}$ and $l \neq i$, consider the following:

$$\hat{P}(X_i = x_i | X_l = x_l) = \sum_{X_{-i-\{l\}}} \hat{P}(X_i = x_i, X_{-i-\{l\}} | X_l = x_l)$$

$$= \sum_{X_{-i-\{l\}}} \hat{P}(X_i = x_i | X_{-i-\{l\}}, X_l = x_l) \hat{P}(X_{-i-\{l\}} | X_l = x_l)$$

$$= \sum_{X_{-i-\{l\}}} \hat{P}(X_i = x_i | X_{-i}) \hat{P}(X_{-i-\{l\}} | X_l = x_l)$$

Node $i$ is rank 2 with respect to $\hat{P}$ and $\mathrm{MB}_G(i)$

$$= \sum_{X_{-i-\{l\}}} (Q_i(X_i = x_i) + \sum_{j \in -i} Q_{ij}(X_i = x_i, X_j)) \hat{P}(X_{-i-\{l\}} | X_l = x_l)$$

$$= Q_i(X_i = x_i) + \sum_{\substack{j \in -i \\ j \neq l}} \sum_{X_j} Q_{ij}(X_i = x_i, X_j) \hat{P}(X_j | X_l = x_l)$$

$$+ Q_{il}(X_i = x_i, X_l = x_l)$$

Now $\hat{P} \in \mathbb{P}$

$$\mathcal{P}(X_i = x_i | X_l = x_l) = Q_i(X_i = x_i) + \sum_{\substack{j \in -i \\ j \neq l}} \sum_{X_j} Q_{ij}(X_i = x_i, X_j) \mathcal{P}(X_j | X_l = x_l)$$

$$+ Q_{il}(X_i = x_i, X_l = x_l)$$

$$\mathcal{P}(X_i = x_i, X_l = x_l) = Q_i(X_i = x_i) \mathcal{P}(X_l = x_l) + \sum_{\substack{j \in -i \\ j \neq l}} \sum_{X_j} Q_{ij}(X_i = x_i, X_j) \mathcal{P}(X_j, X_l = x_l)$$

$$+ Q_{il}(X_i = x_i, X_l = x_l) \mathcal{P}(X_l = x_l) \tag{21}$$

We only focus on the case when $x_i = 0$ because that would be sufficient to determine the Markov Blanket for node $i$. Equation (21) may not have a unique solution because for any pair of nodes $i, j$ if $Q_i(X_i = 0)$, $Q_{ij}(X_i = 0, X_j = 0)$ and $Q_{ij}(X_i = 0, X_j = 1)$ are part of a solution then there exists a solution with $Q_i(X_i = 0) + \epsilon$, $Q_{ij}(X_i = 0, X_j = 0) - \epsilon$ and $Q_{ij}(X_i = 0, X_j = 1) - \epsilon$. We focus on a particular solution where $\tilde{Q}_i(X_i = 0) = Q_i(X_i = 0) + \sum_{j \in \mathrm{MB}_G(i)} Q_{ij}(X_i = 0, X_j = 1)$, $\tilde{Q}_{ij}(X_i = 0, X_j = 0) = Q_{ij}(X_i = 0, X_j = 0) - Q_{ij}(X_i = 0, X_j = 1)$ and thus equation (21) becomes:

$$\mathcal{P}(X_i = 0, X_l = x_l) = \tilde{Q}_i(X_i = 0) \mathcal{P}(X_l = x_l) + \sum_{\substack{j \in -i \\ j \neq l}} \tilde{Q}_{ij}(X_i = 0, X_j = 0) \mathcal{P}(X_j = 0, X_l = x_l)$$

$$+ \tilde{Q}_{il}(X_i = 0, X_l = x_l) \mathcal{P}(X_l = x_l), \ \forall \, l = \{1, \ldots, n\}, l \neq i, x_l \in \{0, 1\} \tag{22}$$

Equation (22) can be written as a system of linear equations:

$$\mathbf{y} = \mathbf{A}\mathbf{q} \tag{23}$$

where $\mathbf{y} \in \mathbb{R}^{2n-2}, \mathbf{A} \in \mathbb{R}^{2n-2 \times n}$ and $\mathbf{q} \in \mathbb{R}^n$. We define $\mathbf{q}$ as follows:

$$\mathbf{q}_j = \begin{cases} \tilde{Q}_{ij}(X_i = 0, X_j = 0), & \text{if } j < i \\ \tilde{Q}_{ij+1}(X_i = 0, X_{j+1} = 0), & \text{if } i \leqslant j \leqslant n-1 \quad \forall j \in \{1, \ldots, n\} \\ \tilde{Q}_i(X_i = 0), & \text{if } j = n \end{cases} \tag{24}$$

The rows of $\mathbf{y}$ and $\mathbf{A}$ are indexed by $l$ and $X_l$, i.e.,

$$\mathbf{y}(l, X_l = x_l) = \mathcal{P}(X_i = 0, X_l = x_l)$$

$$\mathbf{A}(l, X_l = x_l) = \begin{cases} \mathcal{P}(X_j = 0, X_l = x_l), & \text{if } j < i \\ \mathcal{P}(X_{j+1} = 0, X_l = x_l), & \text{if } i \leqslant j \leqslant n-1 \\ \mathcal{P}(X_l = x_l), & \text{if } j = n \end{cases} \tag{25}$$

We take $\mathcal{P}(X_l = 0, X_l = 0) = \mathcal{P}(X_l = 0)$ and $\mathcal{P}(X_l = 0, X_l = 1) = 0$. We can remove the linearly dependent rows from the above system of equations. For simplicity, let us assume that $i = n$. Then for $l = \{2, \ldots, n-1\}$, $\mathbf{y}(1, X_1 = 0) + \mathbf{y}(1, X_1 = 1) - \mathbf{y}(l, X_l = 0) = [\mathbf{A}(1, X_1 = 0) + \mathbf{A}(1, X_1 = 1) - \mathbf{A}(l, X_l = 0)]\mathbf{q}$ is equivalent to $\mathbf{y}(l, X_l = 1) = \mathbf{A}(l, X_l = 1)\mathbf{q}$. Thus we can remove all the rows of $\mathbf{y}$ and $\mathbf{A}$ indexed by $l, X_l = 1, \forall l = \{2, \ldots, n-1\}$ and replace the last row of $\mathbf{y}$ and $\mathbf{A}$ by $\mathbf{y}(1, X_1 = 0) + \mathbf{y}(1, X_1 = 1)$ and $\mathbf{A}(1, X_1 = 0) + \mathbf{A}(1, X_1 = 1)$ respectively. A similar argument can be presented for the case when $i \neq n$. $\qquad \square$

## A.5 Proof of Lemma 1

**Lemma 1** *The population matrix $\overline{\mathbf{A}}$ as defined in equation (14) is a positive semidefinite matrix.*

*Proof.* Here we carry out the proof for $i = n$. The same argument can be applied when $i \neq n$. Consider a random vector $\mathbf{z} \in \mathbb{R}^n$ such that $z_j = \mathbf{1}[X_j = 0], \forall j = \{1, \ldots, n-1\}$ and $z_n = 1$. Note that $\mathcal{P}(X_i = 0) = \mathbb{E}[\mathbf{1}[X_i = 0]] = \mathbb{E}[\mathbf{1}[X_i = 0]^2]$ and $\mathcal{P}(X_i = 0, X_j = 0) = \mathbb{E}[\mathbf{1}(X_i = 0)\mathbf{1}(X_j = 0)], \forall i, j \in \{1, \ldots, n\}$. Thus $\overline{\mathbf{A}} = \mathbb{E}[\mathbf{z}\mathbf{z}^\mathsf{T}]$ which is a positive semidefinite matrix. $\qquad \square$

## A.6 Proof of Lemma 2

**Lemma 2** *If $\tilde{Q}_{ij}(\cdot, \cdot), \forall j \in \{1, \ldots, n\}, j \neq i$ is computed by solving system of linear equations (14) and $\hat{P} \in \mathbb{P}$ is faithful to G then $\tilde{Q}_{ij}(\cdot, \cdot) \neq 0, \forall j \in \{1, \ldots, n\}, j \neq i$ if and only if $j \in \mathrm{MB}_G(i)$.*

*Proof.* For the first part, suppose $\exists j \notin \mathrm{MB}_G(i)$ for which $\tilde{Q}_{ij}(\cdot, \cdot) \neq 0$, then expanding $\hat{P}(X_i | X_{-i}) = \tilde{Q}_i(\cdot) + \sum_{j=1, j \neq i}^n \tilde{Q}_{ij}(\cdot, \cdot)$, we see that $\hat{P}(X_i | X_{-i}) \neq \hat{P}(X_i | X_{\mathrm{MB}_G(i)})$ which violates the faithfulness assumption. For the reverse, suppose $\exists j \in \mathrm{MB}_G(i)$ for which $\tilde{Q}_{ij}(\cdot, \cdot) = 0$. This implies that $X_i$ and $X_j$ are independent given all the other nodes which again violates faithfulness. $\qquad \square$

## A.7 Proof of Lemma 3

**Lemma 3** *$N = \mathcal{O}(\frac{\log n}{\epsilon^2})$ i.i.d observations are sufficient to measure elements of $\overline{\mathbf{A}}$ and $\overline{\mathbf{y}}$, $\epsilon$ close to their true value. That is $|\overline{\mathbf{A}} - \hat{\mathbf{A}}| \leqslant \epsilon$ and $|\overline{\mathbf{y}} - \hat{\mathbf{y}}| \leqslant \epsilon$, for some $\epsilon > 0$ with probability at least $1 - 2\exp(\log(\binom{n}{2} + 3n) - \frac{N\epsilon^2}{2})$ for some $\epsilon > 0$ where $\hat{\mathbf{A}}$ and $\hat{\mathbf{y}}$ are the empirical measurements of $\overline{\mathbf{A}}$ and $\overline{\mathbf{y}}$ respectively and $|\cdot - \cdot|$ denotes componentwise comparison for matrices.*

*Proof.* For the observational data, we need to infer $\binom{n}{2}$ probabilities of the form $P(X_i = 0, X_j = 0), \forall i, j \in \{1, \ldots, n\}$, $n$ probabilities of the form $P(X_i = 0), \forall i = \{1, \ldots, n\}$, $n$ probabilities each of the form $P(X_i = 0, X_1 = 1)$ and $P(X_i = 0, X_2 = 1), \forall i = \{1, \ldots, n\}$. Considering some ordering for $(X_i = x_i, X_j = x_j) = x_{ij}$. We consider $x_{ij} \leqslant x'_{ij}$ if $x_{ij}$ comes before $x'_{ij}$ in the

ordering. Correspondingly, we can define the CDF $F_{ij}(x_{ij}) \triangleq \mathbb{P}((X_i, X_j) \leqslant x_{ij})$. Now, we can apply Dvoretzky-Kiefer-Wolfowitz inequality(Dvoretzky et al., 1956),

$$\mathbb{P}(\sup_{x_{ij}} |\hat{F}_{ij}(x_{ij}) - F_{ij}(x_{ij})| \geqslant \frac{\epsilon}{2}) \leqslant 2\exp(-\frac{N\epsilon^2}{2}), \forall \epsilon > 0 \tag{26}$$

A similar equation can be written for the CDF of $P(X_i)$:

$$\mathbb{P}(\sup_{x_i} |\hat{F}_i(x_i) - F_i(x_i)| \geqslant \frac{\epsilon}{2}) \leqslant 2\exp(-\frac{N\epsilon^2}{2}), \forall \epsilon > 0 \tag{27}$$

where $N$ is number of i.i.d. samples. We compute actual probabilities by using the CDFs. For example:

$$\sup_{x_i} |\hat{\mathbb{P}}(X_i = x_i) - \mathbb{P}(X_i = x_i)| = \sup_{x_i} |\hat{F}_i(x_i) - \hat{F}_i(x_i - 1) - F_i(x_i) + F_i(x_i - 1)|$$

$$\leqslant \sup_{x_i} |\hat{F}_i(x_i) - F_i(x_i)| + \sup_{x_i} |\hat{F}_i(x_i - 1) - F_i(x_i - 1)|$$

$$\leqslant \epsilon$$

We need to ensure that this happens across all possible computations of probabilities. Thus taking a union bound,

$$\mathbb{P}((\exists X_i) \sup_x |\hat{F}_i(x_i) - F_i(x_i)| \geqslant \frac{\epsilon}{2} \vee (\exists X_i, X_j) \sup_{x_{ij}} |\hat{F}_{ij}(x_{ij}) - F_{ij}(x_{ij})| \geqslant \frac{\epsilon}{2}) \tag{28}$$

$$\leqslant 4\exp(\log(\binom{n}{2} + 3n) - \frac{N\epsilon^2}{2}), \forall \epsilon > 0 \tag{29}$$

$\square$

## A.8 Proof of Lemma 4

**Lemma 4** *Let $\hat{A}$ and $\hat{y}$ be the empirical measurements of $\overline{A}$ and $\overline{y}$ as defined in equation* (14) *respectively such that $|\hat{A} - \overline{A}| \leqslant \epsilon$ and $|\hat{y} - \overline{y}| \leqslant \epsilon$ for some $\epsilon > 0$, where $|\cdot - \cdot|$ denotes componentwise comparison for matrices. Let $\hat{q}$ be the solution to the system of linear equations given by $\hat{y} = \hat{A}\hat{q}$ and $\eta\kappa_\infty(\overline{A}) \leqslant 1$, then $\hat{q}$ recovers $q$ up to signs as long as $N = \mathcal{O}(n)$ i.i.d. measurements are used to measure $\hat{A}$ and $\frac{\max_i |q_i|}{\min_i |q_i|} \leqslant \frac{1 - \eta\kappa_\infty(\overline{A})}{4\eta\kappa_\infty(\overline{A})}$, where $\kappa_\infty(\overline{A}) \triangleq \|\overline{A}\|_\infty \|\overline{A}^{-1}\|_\infty$ is the condition number of $\overline{A}$ and $\eta = \max(\frac{n\epsilon}{\sum_{j=1}^{n-1} \mathcal{P}(X_j=0)+1}, \frac{\epsilon}{\mathcal{P}(X_n=0)})$.*

*Proof.* Note that $\hat{A} > 0$ as long as $N = \mathcal{O}(n)$ Anderson (1962). Here we carry out the proof for node $n$ but similar arguments hold for other nodes as well. First note that We denote $\Delta A \triangleq \hat{A} - \overline{A}$ and $\Delta y \triangleq \hat{y} - \overline{y}$. First note that, $\|\overline{A}\|_\infty = \sum_{j=1}^{n-1} \mathcal{P}(X_j = 0) + 1$ and $\|\overline{y}\|_\infty = \mathcal{P}(X_n = 0)$. Thus, $\|\Delta A\|_\infty \leqslant n\epsilon \leqslant \eta\|\overline{A}\|_\infty = \eta(\sum_{j=1}^{n-1} \mathcal{P}(X_j = 0) + 1)$ and $\|\Delta y\|_\infty \leqslant \epsilon \leqslant \eta\|\overline{y}\|_\infty = \eta\mathcal{P}(X_n = 0)$ for $\eta = \max(\frac{n\epsilon}{\sum_{j=1}^{n-1} \mathcal{P}(X_j=0)+1}, \frac{\epsilon}{\mathcal{P}(X_n=0)})$. Thus, we can invoke Theorem 2.2 from Higham (1994) and write,

$$\frac{\|\hat{q} - q\|_\infty}{\|q\|_\infty} \leqslant \frac{2\eta\kappa_\infty(\overline{A})}{1 - \eta\kappa_\infty(\overline{A})}$$

$$\|\hat{q} - q\|_\infty \leqslant \frac{2\eta\kappa_\infty(\overline{A})}{1 - \eta\kappa_\infty(\overline{A})} \|q\|_\infty \tag{30}$$

It follows that if $\frac{\max_i |q_i|}{\min_i |q_i|} \leqslant \frac{1 - \eta\kappa_\infty(\overline{A})}{4\eta\kappa_\infty(\overline{A})}$ then we recover **q** up to correct signs. $\square$

# B   Sample and Time Complexity without access to any observational data

**Sample Complexity.**   Using the Dvoretzky-Kiefer-Wolfowitz (DKW) inequality(Dvoretzky et al., 1956) for each query independently and then taking the union bound across $m_i$ such queries, we get that each query is at max $\epsilon$ away from its true conditional probability with a probability of at least $1 - \sum_{i=1}^{m_i} 4\exp(-\frac{N_i \epsilon^2}{2})$. Let $N_{\min} \triangleq \min_{i=\{1,\cdots m_i\}} N_i$ be the minimum number of sample we need across all query. Then we need $N_{\min} = \mathcal{O}(\frac{\log m_i}{\epsilon^2})$ samples for each query to estimate probabilities of the form $\mathcal{P}(X_i | X_{\bar{i}} = x_{\bar{i}})$, $\epsilon$ close to the true value with probability at least $1 - 4\exp(\log m_i - \frac{N_{\min}\epsilon^2}{2})$. The black-box outputs observational data for each of our queries independently and thus it needs to output a total of $\mathcal{O}(\max(\frac{nk^3 \log^4 n}{\epsilon^2}(\log k + \log\log n), \frac{nk^3}{\epsilon^2}\log\frac{1}{\delta}(\log k + \log\log n))$ samples.

**Time Complexity.**   Each optimization problem is solved using the logarithmic barrier method which takes $\mathcal{O}(n^3 \sqrt{n} \log n)$ time. This needs to be repeated $\mathcal{O}(nk)$ times. Thus, total time complexity is $\mathcal{O}(n^4 k \sqrt{n} \log n)$.

# C   Sample and Time Complexity with access to some observational data

Regarding the black-box queries, we provide the same argument as Appendix B but for computing $\mathcal{P}(X_i | X_{\text{MB}_G(i)} = x_{\text{MB}_G(i)})$, $\epsilon$ close to the true value with probability at least $1 - 4\exp(\log m_i - \frac{N_{\min}\epsilon^2}{2})$. We need to generate samples for each of our queries independently and thus need a total of $\mathcal{O}(\max(\frac{nk^3 \log^5 k}{\epsilon^2}, \frac{nk^3}{\epsilon^2}\log\frac{1}{\delta}\log k))$ samples.

**Time Complexity.**   For the observational data, we are solving an optimization problem by computing inverse of a $\mathbb{R}^{n \times n}$ matrix and then multiplying it by a $\mathbb{R}^n$ vector. This can be done in $\mathcal{O}(n^3)$ time. We repeat this process for each node, and thus it takes $\mathcal{O}(n^4)$ time. All the inference can be done by only one traversal of the samples. Thus the total time complexity remains $\mathcal{O}(n^4)$.

Regarding the black-box queries, each optimization problem is solved using the logarithmic barrier method which takes $\mathcal{O}(k^3 \sqrt{k} \log k)$ time. This needs to be repeated $\mathcal{O}(nk)$ times. Thus, the total time complexity is $\mathcal{O}(nk^4 \sqrt{k} \log k)$.

# D   Synthetic Experiments

We conducted computational experiments on synthetic data to validate our results. In this section, we report the average performance across 5 independently generated Bayesian networks.

**Generating Bayesian Networks.**   We generated 5 synthetic Bayesian networks on 20 nodes. We first chose a causal order for the nodes. We then generated CPTs for the nodes by making sure that each node's CPT is rank 2 with respect to its parents. The parameters $Q_{ij}(\cdot, \cdot)$ as described in Equation (1) were chosen uniformly at random from $[0, 1]$ while making sure that the resulting DAG is faithful. An example of a Bayesian network is shown in Figure 1.

**Black-box.**   We defined a black-box which can answer conditional probabilities queries $BB(i, A, x_A, N)$ to compute $\mathcal{P}(X_i | X_A = x_A), \forall A \subseteq \{1, \ldots, n\}$. The black-box outputs $N$ i.i.d. samples for $X_i$ given $X_A = x_A$.

## D.1   Recovering DAG without Access to any Observational Data.

For the first set of experiments, we did not have access to any observational data. Algorithm 2 takes a Bayesian network on $S \subseteq \{1, \ldots, n\}$ nodes and outputs terminal nodes $T \subseteq S$. The iterative use of Algorithm 2 in Algorithm 1, subsequently provides the exact DAG. We assume that the second node in the causal order does not have any parents. Following Theorem 3, we submit $m_i = 10^C \max(k^2 \log^4 n', k^2 \log 1/\delta)$ queries for each node $i$ at each iteration where $k$ is the maximum number of nodes in Markov blanket, $n'$ is number of nodes in the Bayesian network at a specific iteration, i.e., $n' = |S|$ and $C$ is the control parameter. We fixed $k = 4$ and $\delta = 0.01$. The

Figure 1: An example of synthetic Bayesian network generated on $n = 20$ nodes

number of queries was capped at $300$ to ensure that we do not end up making too many queries. For each query, we only had access to $N = \mathcal{O}(\frac{\log m_i}{\epsilon^2})$ samples from the black-box.

**Results.** We measured the performance of our method by measuring the Hamming distance between the true DAG and the recovered DAG. We also measured recall and precision for our method and then computed the $F1$ score to see their joint effect. The performance measures are defined formally as:

$$\text{Hamming Distance} = \sum_{i=1}^{n}(|\hat{\pi}(i)\backslash\pi_G(i)| + |\pi_G(i)\backslash\hat{\pi}(i)|)$$

$$\text{Precision} = \frac{\sum_{i=1}^{n}|\hat{\pi}(i) \cap \pi_G(i)|}{\sum_{i=1}^{n}|\hat{\pi}(i)|}$$

$$\text{Recall} = \frac{\sum_{i=1}^{n}|\hat{\pi}(i) \cap \pi_G(i)|}{\sum_{i=1}^{n}|\pi_G(i)|}$$

$$\text{F1 Score} = \frac{2 \times \text{Precision} \times \text{Recall}}{\text{Precision} + \text{Recall}}$$

where $\pi_G(i)$ is the set of true parents of node $i$ in true DAG $G$ and $\hat{\pi}(i)$ is the recovered set of parents of node $i$. Note that the recovery of a reversed edge is treated as a mistake. We show the average performance of our method across $5$ independently generated Bayesian networks.

Observe that in Figure 2a the Hamming distance goes towards zero as we increase the number of samples, or equivalently, as we increase the control parameter $C$. Similarly, in Figure 2c, 2d both precision and recall (and $F1$ score as a result in Figure 2b) go towards $1$ as we increase the control parameter $C$ in our experiments with a sharp transition around $C = -1$. This is consistent with our expected results from Theorem 3 and validates our theory.

### D.2 Recovering Markov Blanket with Access to Some Observational Data.

For the second set of experiments, we had access to some observational data. Our method can be made more efficient by first computing the Markov blanket for a node and then applying Algorithm 2 with queries of the form $f_i(A_j) = \mathcal{P}(X_i \mid X_{A \cap \text{MB}_G(i)} = x_{A \cap \text{MB}_G(i)})$. Since, usually $|A| \gg |A \cap \text{MB}_G(i)|$, this saves a lot of computational efforts and Black-box queries for our algorithm. Note that $n$ observations are necessary for Lemma 4 to work. Beyond this, from Lemma 3, we only

(a) Hamming distance with control parameter $C$

(b) $F1$ score with control parameter $C$

(c) Precision with control parameter $C$

(d) Recall with control parameter $C$

Figure 2: Regime without observational data. Plots of Hamming distance, $F1$ score, precision and recall versus the control parameter $C$ for Bayesian networks on $n = 20$ nodes with $m_i = 10^C \max(k^2 \log^4 n', k^2 \log 1/\delta)$ queries for each node $i$.

require $\mathcal{O}(\frac{\log n}{\epsilon^2})$ observational samples for recovering the Markov blankets of all the nodes. Thus, we conducted the experiments by generating $N = \max(10^C \frac{\log n}{\epsilon^2}, n)$ observational samples. The results of the experiments are provided below.

**Results.** As before, we measured performance of our method by measuring the Hamming distance between the true Markov blankets and the recovered ones. We also measured recall and precision for our method and then computed the $F_1$ score to see their joint effect. The performance measures are defined slightly differently as the recovery is with respect to the Markov blankets.

$$\text{Hamming Distance} = \sum_{i=1}^{n} (|\hat{\text{MB}}(i) \setminus \text{MB}_G(i)| + |\text{MB}_G(i) \setminus \hat{\text{MB}}(i)|)$$

$$\text{Precision} = \frac{\sum_{i=1}^{n} |\hat{\text{MB}}(i) \cap \text{MB}_G(i)|}{\sum_{i=1}^{n} |\hat{\text{MB}}(i)|}$$

$$\text{Recall} = \frac{\sum_{i=1}^{n} |\hat{\text{MB}}(i) \cap \text{MB}_G(i)|}{\sum_{i=1}^{n} |\text{MB}_G(i)|}$$

$$\text{F1 Score} = \frac{2 \times \text{Precision} \times \text{Recall}}{\text{Precision} + \text{Recall}}$$

where $\text{MB}_G(i)$ is the set of nodes in the Markov blanket of node $i$ in true DAG $G$ and $\hat{\text{MB}}(i)$ is the recovered set of nodes in the Markov blanket of node $i$. Below we provide average performance of our method across 5 independently generated Bayesian networks.

We see in Figure 3a that the Hamming distance of Markov blanket recovery goes to zero as we increase number of observational samples, or equivalently, as we increase the control parameter $C$. Similarly, precision and recall of Markov blanket recovery in Figure 3c, 3d approach 1 as number

(a) Hamming distance of Markov blanket recovery with control parameter $C$

(b) $F1$ score of Markov blanket recovery with control parameter $C$

(c) Precision of Markov blanket recovery with control parameter $C$

(d) Recall of Markov blanket recovery with control parameter $C$

Figure 3: Regime with observational data. Plots of Hamming distance, $F1$ score, precision and recall for Markov blanket recovery versus the control parameter $C$ for Bayesian networks on $n = 20$ nodes with $N = \max(10^C \frac{\log n}{\epsilon^2}, n))$ observational samples

of observational samples increase. This validates our theory. Another interesting observation is that recall is very close to 1 even for a small number of observational samples. This is good for our method as it would still work when recovering any set $S$ such that $\mathrm{MB}_G(i) \subseteq S$. The sample and time complexities are improved depending on the size of $S$ (the best result is achieved when $S = \mathrm{MB}_G(i)$).

After we recovered the Markov blanket, we executed our Algorithm 2 with $f_i(A_j) = \mathcal{P}(X_i \mid X_{A \cap \mathrm{MB}_G(i)} = x_{A \cap \mathrm{MB}_G(i)})$. We then obtained similar results as in the no-observational-data regime, but with smaller number of samples and less computation.