[Reviews · NeurIPS 2019]

Reviewer 1



To summarize the basic idea: by assuming a low-rank representation of CPTs, it becomes possible to identify the leaves, and their parents. One can then iteratively learn the structure from leaves to roots. The low-rank representation of a CPT could be viewed as a mixture of a set of CPTs with a bounded number of parents. The paper would be improved with some experiments that support the claims. There are some preliminary results provided in the supplementary material, but the plots are noisy and don't really support well the claim that the algorithm recovers the true structure as the amount of data/queries increases. On the plots: rather than vary C on the x-axis, it would be more straightforward to plot the amount of data/queries, which would be more direct and easier to interpret.

Reviewer 2



The paper studies the problem of learning Bayesian network structures from data under a particular setting as specified under Assumptions 1 and 3 and assuming all variables are binary. It presents an algorithm that correctly recovers the true structure under Assumption 4, and provides the sample and time complexity of the algorithm. Overall the paper is clearly written. The results are nice under the setting studied in the paper. My main concern is with the motivation of the problem setting. Why is this problem setting (Assumptions 1 and 3) interesting or useful in practice? What does a CPT being rank 2 or low rank mean in practice? The paper claims that the proposed method can be easily extended for any rank k CPTs. Can you provide a discussion of how this can be done? I wonder whether Assumption 4 becomes less likely to hold when k becomes larger. How strong is Assumption 4? Under what situations will Assumption 4 be violated? Perhaps an example situation where Assumption 4 is violated will be helpful. The paper assumes all variables are binary. Are the results extendable to any discrete variables? -line 35, the citation entry (D., 1996) should be fixed. -line 195, “unifromly” -> uniformly ---------------- Comments after author response: Thanks for providing a motivation to the problem setting. I still have concerns over Assumption 4. It looks to me Assumption 4 could easily be violated. I wonder its impact on the correctness of the algorithm.

Reviewer 3



This paper presents a method for structural learning of a BN given observational data. The work is mainly theoretical, and for the proposal some assumptions are taken. A great effort is also given in presenting and develop theoretically the complexity of the algorithm. One of the key points in the proposed algorithm is the use of Fourier basis vectors (coefficients) and how they are applied in the compressed sensing step. I haven't checked thoroughly all the mathematical part, which is the core of the paper. Being familiarised with BNs but not with the Fourier transformation, this part seems hard to be evaluated from my side. However, the theory seems to have shown and extensively stated (also in the supplementary material). My biggest concern is the restriction of only two parents for a node. I am aware that authors indicate it wouldn't be a problem to make this number (rank) larger, but then one wonders why not using a more realistic approach. I can see that the paper is mainly theoretical and I wouldn't say this is not enough if the methodology is correctly presented, as this is the case. Anyway, in a so practical issue as learning BNs one would expect some empirical evaluation with at least a classical benchmark of datasets. You could have used some 'simulation' for the black-box solved queries. All in all, I like the idea, the way the work is presented, and I think this paper could generate interesting debate in the NeurIPS community. Minor comments: There are some places were English could be improved, so I suggested further checking. For instance: - Introduction, motivation: Bayesian networks are one of the most important classES of probabilistic graphical models - Introduction, related work: "... data is a well known but AN incredibly difficult problem to solve in the machine learning community"

[Author Response · NeurIPS 2019]

1. **[ALL]** As R3 appreciates, our paper is mainly theoretical in nature and the focus has been to present a correct and theoretically sound methodology. We believe that *the beauty of our theory* is that we establish a connection between the notion of rank-2 succinct representation, and Fourier transformation of a set function. We show that Fourier coefficients for sets of size 3 or bigger are 0. Moreover, for terminal nodes only singleton sets of the parents have non-zero Fourier coefficients. This connection not only allows us to differentiate between terminal and non-terminal nodes but also enables us to identify parents of terminal nodes.

2. **[R1]** Regarding "plots are noisy and don't really support well the claim that the algorithm recovers the true structure as the amount of data/queries increases" - The experiments provided in our paper validate our theory. Check the sharp jump in Figure 2 which is expected based on Theorem 3. Similarly, Figure 3 shows that Markov blanket can be recovered with sufficient number of observational data. Some variance in the plots is expected as experiments are conducted for multiple networks. In point 8, we provide more experimental evidence to further validate our theoretical contribution.

3. **[R2] On Motivation.** Learning structure of Bayesian network from data, in its general form, is provably NP-hard [Chickering, 1996, Learning Bayesian Networks Is NP-Complete]. If P$\neq$NP, then it is absolutely necessary to exploit further structure of NP-hard problems to solve them in polynomial time and samples. For instance, consider succinctness assumptions in other problems: low rankness in matrix completion or sparsity in compressive sensing. We consider Bayesian network with nodes having potentially complicated probability tables which can be succinctly represented as a sum of small and less complicated probability tables (Lines 52-57, 100-108 ). We consider the case where the smaller tables depend only on the node and one parent (*rank*-2 case). Assumption 3 ensures that such a succinct representation is possible (check point 4 for a practical example). Assumption 1 is reminiscent of (but not equivalent to) the interventional setting (Lines 130-132). Assumption 1 could be seen as an interactive query and in special settings (but not always), one could think of it as expert knowledge.

4. **[R2] A particular example used in practice.** The *combinational stochastic logic gates*[Mansinghka et al, 2008, Stochastic digital circuits for probabilistic inference] are heavily used in digital hardware. Consider this simple $\Theta$-gate which can be easily represented as a rank-2 CPT, i.e., $P(Z|X,Y) = Q_z(Z) + Q_{zx}(Z,X) + Q_{zy}(Z,Y)$ where $Q_z(Z) = 0, \forall Z \in \{0,1\}$ and tables $Q_{zx}$ and $Q_{zy}$ are shown below.

| $X$ | $Y$ | $P(Z=0 \mid X,Y)$ | $P(Z=1 \mid X,Y)$ |
|---|---|---|---|
| 0 | 0 | 1 | 0 |
| 1 | 0 | 0.5 | 0.5 |
| 0 | 1 | 0.5 | 0.5 |
| 1 | 1 | 0 | 1 |

(a) $\Theta$-gate truth table

| | $X=0$ | $X=1$ |
|---|---|---|
| $Z=0$ | 0.5 | 0 |
| $Z=1$ | 0 | 0.5 |

(b) $Q_{zx}$

| | $Y=0$ | $Y=1$ |
|---|---|---|
| $Z=0$ | 0.5 | 0 |
| $Z=1$ | 0 | 0.5 |

(c) $Q_{zy}$

5. **[R2, R3]** Our theory works for any general rank-$k$ CPTs. Rank-2 is only used for clarity. For rank-$k$ CPTs, we differentiate terminal and non-terminal nodes by looking at non-zero terms in $\hat{f}_i(B)$ for some $|B| = k$ (Assumptions 3, 4 and Theorems 1, 2, 3 are updated accordingly).

6. **[R2]** Reviewer 2 has asked to present a case where Assumption 4 is violated. Assumption 4 does not hold only when non-terminal nodes are rank-2 (or rank-k) with respect to their Markov blankets (Take $A(i) = \texttt{MB}(i)$ in Eq (1)). This condition may occur for a node with just one parent and one child (and no other parent of the child).

7. **[R2] Extending theory to discrete variables.** We chose binary variables for ease of presentation, our results easily extend to discrete variables. The most crucial part of the theory is to map $\mathcal{P}(X_r = x_r | X_{\bar{r}} = x_{\bar{r}})$ to a set function. Assume that every variable can take 4 values. We encode them as: $00, 01, 10, 11$. We choose a set $S \subseteq \{1, \cdots, 2n\}$ and assign variable $X_i$ a 2-bit value $x_i$. The first bit of $x_i$ is 1 if $2i - 1 \in S$ and the second bit of $x_i$ is 1 if $2i \in S$. The rest of the theory follows once we have this map in place.

8. **[R1, R3] Experimental results. 1.** We see a trend similar to Figure 2 of our paper for bigger networks.

| Control Parameter | # Queries | Precision | Recall | Control Parameter | # Queries | Precision | Recall |
|---|---|---|---|---|---|---|---|
| -2 | 37 | 55 % | 82 % | -2 | 71 | 84% | 96% |
| -1.5 | 118 | 91% | 96 % | -1.5 | 227 | 89% | 97% |
| -1 | 374 | 93 % | 93 % | -1 | 719 | 99.5% | 99 % |

(a) $n = 50$ nodes      (b) $n = 100$ nodes

**2.** A baseline comparison: on 20 nodes network, our method (precision 95%, recall 95%) with 300 queries outperforms state-of-the-art MMHC method (precision 86%, recall 86%) and greedy (precision 80%, recall 82%) with 100000 samples. **3.** Using recovered Markov blanket (Figure 3, 20 nodes), we can recover DAG with 95% precision and 95% recall.

9. **[R2, R3]** Formatting errors will be corrected in the final version.

[Meta-Review · NeurIPS 2019]

The paper introduces a notion of low rank conditional probability tables (CPTs). and presents a method for learning Bayesian network structures with low rank CPTs. Overall the reviewers found the results innovative, interesting and theoretically justified. Many of the reviewers concerns were properly addressed in the rebuttal. Several reviewers stated that more empirical work could have greatly benefit the paper.